# Risk Factors for Hypoparathyroidism after Thyroid Surgery: A Single-Center Study

**DOI:** 10.3390/jcm12051956

**Published:** 2023-03-01

**Authors:** Francesca Privitera, Danilo Centonze, Sandro La Vignera, Rosita Angela Condorelli, Costanza Distefano, Rossella Gioco, Alba Ilari Civit, Giordana Riccioli, Ludovica Stella, Andrea Cavallaro, Matteo Angelo Cannizzaro, Massimiliano Veroux

**Affiliations:** 1Department of General Surgery, Azienda Ospedaliera Universitaria Policlinico San Marco, 95123 Catania, Italy; 2Surgery Unit, Enna Hospital, 94100 Enna, Italy; 3Department of Clinical and Experimental Medicine, University of Catania, 95124 Catania, Italy; 4Department of Medical and Surgical Sciences and Advanced Technologies, University Hospital of Catania, 95123 Catania, Italy

**Keywords:** parathyroid hormone, hypocalcemia, thyroid cancer, incidental parathyroidectomy, parathyroid hormone, female, thyroidectomy, lobectomy, central neck dissection, multinodular goiter

## Abstract

Background: Hypoparathyroidism is one of the most common complications after thyroidectomy. This study evaluated the incidence and potential risk factors for postoperative hypoparathyroidism after thyroid surgical procedures in a single high-volume center. Methods: In this retrospective study, in all patients undergoing thyroid surgery from 2018 to 2021, a 6 h postoperative parathyroid hormone level (PTH) was evaluated. Patients were divided into two groups based on 6 h postoperative PTH levels (≤12 and >12 pg/mL). Results: A total of 734 patients were enrolled in this study. Most patients (702, 95.6%) underwent a total thyroidectomy, while 32 patients underwent a lobectomy (4.4%). A total of 230 patients (31.3%) had a postoperative PTH level of <12 pg/mL. Postoperative temporary hypoparathyroidism was more frequently associated with female sex, age < 40 y, neck dissection, the yield of lymph node dissection, and incidental parathyroidectomy. Incidental parathyroidectomy was reported in 122 patients (16.6%) and was correlated with thyroid cancer and neck dissection. Conclusions: Young patients undergoing neck dissection and with incidental parathyroidectomy have the highest risk of postoperative hypoparathyroidism after thyroid surgery. However, incidental parathyroidectomy did not necessarily correlate with postoperative hypocalcemia, suggesting that the pathogenesis of this complication is multifactorial and may include an impaired blood supply to parathyroid glands during thyroid surgery.

## 1. Introduction

Total thyroidectomy is one of the most frequently performed endocrine surgical procedures, and it can lead to potentially serious complications, including hypoparathyroidism [1,2,3]. Hypoparathyroidism is the most frequent complication after total thyroidectomy and may affect 3–49% of patients [1,2,3,4,5,6,7,8,9,10,11,12,13,14,15,16], with an incidence of definitive hypoparathyroidism of 4.11% at 6 months postoperatively [17]. Postoperative low levels of parathyroid hormone (PTH) and consequent hypocalcemia may be associated with an injury to parathyroid glands because of incidental removal of one or more glands, or to a compromised blood supply as a result of their dissection [3,12], and this risk may be further increased in patients undergoing total thyroidectomy and a neck dissection [3,9,11,14,15]. Postoperative hypocalcemia may present with no symptoms or with severe complications such as laryngospasm, tetany, cardiac arrhythmias, and death [4], and hypocalcemia may cause significant morbidity, including a delay in hospital discharge and the need for intravenous calcium supplementation [4,9,18]. Traditionally, serum calcium levels have been used to evaluate for hypocalcemia; however, this can be confounded by the administration of prophylactic calcium during the postoperative period [4]. Because of the very short half-life of PTH, PTH levels may be used as an early predictor of post-thyroidectomy hypocalcemia [1,6,9,10]; however, conflicting data have been reported on the most useful predictive threshold for postoperative PTH levels and the appropriate timing of PTH measurement after surgery. Most authors, however, agreed that PTH levels taken within 1–6 h after skin closure may be predictive of postoperative hypocalcemia [3,4,9,11,12,15,19]. Alternatively, a reduction in postoperative PTH from the first postoperative day (POD1) may be a good alternative [6,20,21].

In this study we have evaluated the risk factors for hypoparathyroidism after thyroid surgery to identify the patient and surgical-related factors that could have a significant correlation with the development of this complication. Moreover, the incidence and the predictive risk factors associated with incidental parathyroidectomy were also evaluated.

## 2. Materials and Methods

All patients who were scheduled for surgical treatment for thyroid disease between January 2018 and December 2021 were retrospectively reviewed. All patients were scheduled for surgery based either on clinical evaluation or fine needle aspiration. Patients with the affection of calcium homeostasis or patients undergoing completion thyroidectomy were excluded from the analysis. Written informed consent detailing all of the surgical procedures and possible complications was obtained from all patients. Surgical techniques and clinical and follow-up data characteristics have been previously described [9]. In brief, all patients underwent a thyroid lobectomy or a thyroidectomy by the same team of two surgeons. Central neck dissection was performed accordingly for the presence of enlarged monolateral or bilateral lymph nodes, as previously described [9]. The biochemical monitoring of mineral homeostasis included serum calcium, phosphate, and PTH, measured preoperatively and 6 h after surgery, as previously described [9].

Patients were divided into two groups based on postoperative PTH levels (≤12 and >12 pg/mL), based on the lowest normal value reported by the in-house laboratory. Thyroid volume was calculated by the ellipsoid formula, as previously described [9]. Statistical data analysis was performed using SPSS (version 20.0; SPSS Inc. Chicago, IL, USA). Data are expressed as mean ± standard deviation (SD). To compare parametric variables, the Pearson chi-square test or Fisher’s exact test was used. To compare nonparametric variables, the Student’s *t*-test or Mann–Whitney U test was used. The difference between the two means was calculated using the Wilson test. Odds ratios (OR) were reported with a 95% confidence interval (95% CI) and *p*-values. The level of statistical significance was determined at *p* < 0.05. Predictive factors of surgical complications with a *p*-value < 0.5 in univariate analysis were considered for the multivariate model using a downward stepwise binary logistic regression analysis.

## 3. Results

A total of 734 patients underwent thyroid surgery between January 2018 and December 2021. There were 573 women (78.0%) and 161 men (22.0%), with a mean age of 53.3 years (range, 18–76). Patients’ demographics, operative details, histological findings, and postoperative events are reported in Table 1.

Most patients (702, 95.6%) underwent a total thyroidectomy, while 32 patients underwent lobectomy (4.4%). Multinodular goiter was the most frequent indication for surgery (64.9%), while 178 (24.2%) patients had thyroid cancer.

Postoperative temporary hypocalcemia was present in 231 (31.4%) patients; however, most patients only had mild symptoms (numbness or tingling). At the 6-month follow-up, 17 (2.3%) patients experienced definitive hypocalcemia.

A total of 230 patients (31.3%) had a postoperative PTH level of <12 pg/mL (Table 2). Female patients < 40 y were at higher risk of low PTH levels after thyroid surgery. The postoperative levels of PTH were not significantly affected by histological diagnosis or thyroid volume, and the presence of thyroid cancer was not related to an increased risk of low PTH levels after thyroid surgery; however, patients with lymph node metastasis were at higher risk of postoperative hypoparathyroidism (5.2%, *p* = 0.028). Interestingly, patients undergoing a neck dissection significantly increased the risk of parathyroid injury (OR = 1.534 (95% CI 1.095–2.148), *p* < 0.05), and a higher mean number of lymph nodes retrieved correlated significantly with the incidence of hypoparathyroidism (8.30 vs. 4.09, *p* = 0.005). While the presence of thyroiditis may, in principle, be associated with a more challenging surgical procedure, it was not correlated with a higher incidence of postoperative hypocalcemia.

Incidental parathyroidectomy was present in 27% of patients with a postoperative PTH level of <12 pg/mL (OR 2.62 (95%CI 1.763–3.897), *p* < 0.001), but 61 patients (12.1%) with normal postoperative PTH levels had an incidental parathyroidectomy, suggesting that even if parathyroids seem well preserved, their function may be impaired. Indeed, as previously reported in patients with thyroid cancer (9), there was no significant correlation between the number of parathyroid glands identified during surgery with the incidence of postoperative hypoparathyroidism.

PTH levels were significantly different between the two groups (4.8 ± 3.73 pg/mL vs. 37.06 ± 17.64 pg/mL, *p* < 0.001). Overall, patients with postoperative hypocalcemia had 65.7% lower PTH levels compared to preoperative levels, and this difference was highly evident in patients with postoperative PTH levels < 12 pg/mL (90.1%).

Temporary postoperative hypocalcemia developed in 147 patients (64%) in the group with PTH levels < 12 pg/mL compared to 84 patients (16.7%, *p* < 0.001) in the group with PTH levels > 12 pg/mL. Interestingly, 17 patients (7%) with a postoperative PTH value of <12 pg/mL developed a definitive hypocalcemia, while no patient in the group with postoperative PTH levels > 12 pg/mL developed a definitive hypoparathyroidism. In multivariate analysis, female sex, neck dissection, incidental parathyroidectomy, and PTH levels < 12 pg/mL were predictive of postoperative temporary hypocalcemia, while age did not relate to an increased risk of hypocalcemia (Table 3). Moreover, a postoperative PTH level of <5 pg/mL significantly increased the risk for definitive hypoparathyroidism (OR = 23.2 (95% CI 2.45–208.43, *p* < 0.0001). At the 6-month follow-up, there were no significant differences in serum calcium or PTH levels among the two groups.

A subsequent analysis of risk factors for incidental parathyroidectomy was performed (Table 4). A total of 122 patients (16.6%) had an incidental parathyroidectomy. Female patients with an age of <55 years had a higher incidence of incidental parathyroidectomy, and younger (<40 years) patients had the highest risk of having an incidental parathyroidectomy (RR 1.53 (95% CI 1.084–2.161), OR 1.72 (95% CI 1.091–2.710), *p* = 0.014), compared to patients > 41 years (OR 0.54 (95 CI 0.402–1.194, *p* = 0.583)). Patients with thyroid cancer were at higher risk of having an incidental parathyroidectomy (RR 1.5 (95 CI 1.096–1.931) OR 1.64 (95% CI 1.116–2.572, *p* = 0.013), and this risk was further increased in patients who underwent neck dissection (RR 1.77 (95% CI 1.293–2.114), OR 2.2 95% CI 1.431–3.197, *p* < 0.001); notably, the risk was also related to the number of retrieved lymph nodes, being the highest for > 8 lymph nodes (OR 3.3 (95% CI 1.26–8.75), *p* = 0.010) retrieved. The risk of incidental parathyroidectomy did not correlate with the number of parathyroid glands identified during surgery. Patients with unintentional parathyroidectomy had significantly lower postoperative PTH levels (18.1 ± 20.0 vs. 29.2 ± 23, pg/mL, *p* < 0.001) and this correlated with a significantly higher incidence of postoperative biochemical hypocalcemia in patients with incidental parathyroidectomy (49.1% vs. 38.1%, *p* = 0.024). Interestingly, thyroid volume was a protective factor against incidental parathyroidectomy and those with a larger thyroid were at lower risk of incidental parathyroidectomy.

In multivariate analysis, age < 40 years, postoperative PTH levels < 12 pg/mL, a diagnosis of thyroid cancer, and neck dissection were predictive of postoperative hypocalcemia, while gender and the extent of surgery did not relate to an increased risk of hypocalcemia (Table 5). Definitive hypocalcemia was more common in patients with incidental parathyroidectomy (4.9% vs. 1.7%, *p* = 0.036), although at a 6-month follow-up there were no significant differences in serum calcium levels.

## 4. Discussion

The real incidence of hypoparathyroidism after thyroid surgery is debatable due to the heterogeneity in diagnosis and identification of this complication, which may affect up to 38% of patients [15]. The early identification and stratification of patients at higher risk of complications after thyroid surgery would provide the opportunity for a custom intervention to prevent symptomatic hypocalcemia, thus reducing the hospital postoperative stay.

After thyroidectomy, the monitoring of PTH and serum calcium levels are the best predictors for identifying hypoparathyroidism and treating the consequent symptomatic hypocalcemia [21]; however, there is a clear lack of consensus about the timing, patient selection, and cut-off points for PTH levels [4].

In their recent meta-analysis, Nagel et al. [22] evaluated the correct definition of postoperative hypoparathyroidism, which should consider the changes in calcium and PTH levels and their interdependence together with the presence of clinical symptoms. After their analysis, the authors suggested that hypoparathyroidism could be defined as the presence of an undetectable or inappropriately low postoperative PTH level in the context of hypocalcemia with or without hypocalcemic symptoms [22].

According to the American Thyroid Association, in most studies, the timing of PTH measurements has ranged from 10 min to 24 h post-thyroidectomy [14], and a postoperative PTH level of <15 pg/mL is usually predictive of hypocalcemia [6,8,9,10,11,13,21,23].

While most authors support the early PTH measurements taken within 4–6 h of surgery as potentially predictive of temporary postoperative hypocalcemia [1,6,7,9,24], the timing of PTH measurement after surgery may vary significantly, ranging from intraoperative, 1–4 h after surgery, or on postoperative day 1 [6]. While it is reasonable that an earlier measurement of PTH levels will lead to early administration of therapeutic calcium and vitamin D medication before clinical symptoms, it may also depend on local specific facilities [22].

In their study, Riordan et al. [9] found that postoperative day 1 PTH levels ≥ 15 pg/mL along with calcium levels ≥ 2.0 mmol/L were associated with a low risk of symptomatic hypocalcemia, thus allowing for the discharge of most patients without calcium supplementation. This was consistent with many other studies in the literature that found that a postoperative day 1 PTH level of >10 pg/mL may be a safe threshold for patient discharge without calcium supplementation [9,21,22,23,24,25].

This study explored the incidence of 6 h postoperative low levels of PTH in a large cohort of patients undergoing thyroid surgery. Postoperative hypocalcemia developed in 231 (31.4%) patients, while a total of 230 patients (31.3%) had a postoperative PTH level of <12 pg/mL; among the 230 patients with PTH levels < 12 pg/mL, 147 patients (64%) developed temporary postoperative hypocalcemia compared to 84 of 504 patients (16.7%, *p* < 0.001) in the group with PTH levels > 12 pg/mL. PTH measurements 6 h after surgery may significantly predict postoperative hypoparathyroidism and hypocalcemia; definitive hypoparathyroidism developed only in the group of patients with postoperative PTH levels < 12 pg/mL (17 patients, 7%), suggesting that a PTH level of >12 pg/mL is associated with a low risk of postoperative symptomatic hypocalcemia and may allow, in principle, a safe patient discharge without calcium supplementation.

To increase the sensibility and the ability of PTH levels to predict the risk of hypoparathyroidism, many groups preferred the measuring of intraoperative or postoperative intact PTH levels drawn at various time points in the early post-thyroidectomy period [10,14,21]. Swift et al. [4] evaluated the most appropriate timing for PTH measurement in a cohort of 124 patients in whom PTH levels were measured preoperatively, at 30 min, and at 6 h postexcision; PTH levels at 30 min and 6 h postsurgery were the most important predictors of hypocalcemia. Similar findings were reported by Schlottman et al. [26] and Lombardi et al. [27], who performed PTH measurements at different successive time points and found that a single PTH level between 4 and 6 h postsurgery may accurately predict postoperative hypocalcemia.

This assumption was further demonstrated by Del Rio et al. [3] who showed that among the 101 patients presenting with serum calcium levels < 7.5 mg/dL, only 49 had PTH values less than 12 pg/mL, whereas the other 52 patients had PTH values within the normal range [6]. In contrast, in our study, a higher incidence of postoperative temporary hypocalcemia was observed in the group of patients with postoperative PTH levels < 12 pg/mL, and there was a significant difference in PTH levels between patients with postoperative hypocalcemia (16.7 ± 5.13 pg/mL) and patients with normocalcemia (32.2 ± 24.4 pg/mL, *p* < 0.001), and overall, patients with postoperative hypocalcemia had 65.7% lower PTH levels compared to preoperative levels, and this difference was even more pronounced in patients with postoperative PTH levels < 12 pg/mL (90.1%).

The mechanism of post-thyroidectomy hypoparathyroidism is probably multifactorial and includes a correct surgical technique, a parathyroid injury, patient gender, incidental parathyroidectomy, and neck dissection [9,14,28].

Conflicting results have been reported in the literature about the influence of age in the development of postoperative hypoparathyroidism. While some studies reported an influence of younger age [9,29], many other studies did not confirm such an association [2,3,14]. In our study, given the higher frequency and susceptibility of the female sex to thyroid pathology, female patients < 40 y were at higher risk of developing postoperative hypocalcemia (OR 1.91, 95% CI 1.281–2.943, *p* < 0.001), and this higher predisposition may be a consequence of the effects of sex steroids on PTH secretion [30], although this association was reported with conflicting results [9,14,21].

The number of parathyroid glands identified during surgery and autotransplanted parathyroids did not influence the rate of postoperative hypocalcemia, as reported in many studies [9,12]. Privitera et al. [9] reported that in patients with thyroid cancer, the rate of postoperative hypocalcemia is strongly correlated with the extent of surgery and particularly with neck dissection. This was confirmed in this study, where, although the diagnosis of thyroid cancer was not associated with an increased risk of hypoparathyroidism, patients with lymph node metastasis (5.2%, *p* = 0.028) undergoing neck dissection were at higher risk of postoperative hypoparathyroidism (OR = 1.534 (95% CI 1.095–2.148), *p* < 0.05). Interestingly, patients with low postoperative PTH levels had a higher mean number of lymph nodes retrieved compared to patients with higher PTH levels (8.30 vs. 4.09, *p* = 0.005). This may be partially explained by the high rate of incidental parathyroidectomy reported in patients with thyroid cancer [9], where a clear distinction between enlarged lymph nodes and parathyroid glands is often challenging. Neck dissection in patients with thyroid cancer has been strongly associated with the risk of hypoparathyroidism [1,2,12,13,31], although it seems correlated to surgeon experience and center volume [32]. Although some studies reported that temporary hypocalcemia may be related to the tumor diameter in female patients [1] and the histological diagnosis of papillary cancer [28], we were not able to confirm this association. A total of 122 patients (16.6%) had an incidental parathyroidectomy—younger (<40 years) patients, and patients with thyroid cancer and undergoing a neck dissection were at higher risk for incidental parathyroidectomy. Again, the yield of lymphadenectomy correlated significantly with the risk of incidental parathyroidectomy, particularly for >8 lymph nodes retrieved.

The incidence of incidental parathyroidectomy in the literature is reported between 4% and 29% [9,32,33,34], and thyroid malignancy and neck dissection, together with the surgeon’s experience, have been identified as the strongest risk factors associated with incidental parathyroidectomy [9,32,33,34]. Surgical technique and surgeon expertise may significantly correlate with incidental parathyroidectomy. In their study, Barrios et al. [29], found that central neck dissection, either prophylactic or therapeutic, but not the yield of lymphadenectomy, increased the risk of incidental parathyroidectomy. However, the surgeon’s experience was related to a lower incidence of incidental parathyroidectomy, suggesting that high-volume centers may perform more extensive thyroid surgery with neck dissection with a reasonable rate of postoperative complications [32]. In principle, the identification of parathyroids during surgery may decrease the risk of incidental parathyroidectomy [35]. However, Riordan et al. [36], in their retrospective study on 511 patients undergoing total thyroidectomy, reported an increased incidence of hypocalcemia in patients in whom a greater number of parathyroids had been identified. This would suggest that an extensive identification of parathyroids, even if they are apparently well preserved, may determine a devascularization of the final branch of blood supply to the parathyroids [32,37], finally leading to hypoparathyroidism. This apparent paradox was further confirmed in our analysis, where, although temporary hypocalcemia was significantly more frequent in patients with incidental parathyroidectomy, 27.9% of patients without incidental parathyroidectomy developed temporary hypocalcemia, and 11 patients (1.7%) developed definitive hypocalcemia, suggesting that even if all glands are well preserved, the normal postoperative parathyroid function is not guaranteed [38].

The main limitation of this study is that the field has been already explored in the literature; however, this is one of the largest single-center series reported in the literature and included the role of incidental parathyroidectomy. Moreover, although the study is retrospective, data are homogenous since surgical procedures were performed by the same surgical team in a high-volume center, and this could reduce the bias caused by different surgeon experiences.

In conclusion, thyroid surgery may be associated with an increased risk of postoperative temporary hypoparathyroidism and hypocalcemia. Young female patients undergoing neck dissection are at higher risk of developing temporary hypoparathyroidism. Incidental parathyroidectomy is associated with a high rate of postoperative hypoparathyroidism, but a large proportion of patients without incidental parathyroidectomy may experience postoperative temporary hypocalcemia, suggesting that a careful surgical technique is recommended for reducing the risk of postoperative complications.

## Figures and Tables

**Table 1 jcm-12-01956-t001:** Patients’ characteristics.

Characteristic	*N* (%)
Gender	
Male	573 (78.0)
Female	161 (22.0)
Mean age (years)	53.3 ± 20.6
Surgical Procedure	
Total thyroidectomy	702 (95.6)
Lobectomy	32 (4.4)
Histological Diagnosis	
Multinodular goiter	477 (64.9)
Hashimoto thyroiditis	116 (15.8)
Thyroid cancer	178 (24.2)
Cancer: Histological type	
Papillary	165 (22.4)
Follicular	46 (6.2)
Other (medullary, anaplastic, and rare tumors)	5 (0.6)
TNM Classification	
T1	120 (16.3)
T2	9 (1.2)
T3	48 (6.5)
T4	1 (0.1)
Lymph node metastasis (N+)	13 (1.7)
Unintentional parathyroidectomy	122 (16.6)
Portion of parathyroid	20 (26.4)
One parathyroid	102 (83.6)
Parathyroid glands identified during surgery	
0	11 (1.5)
1	36 (4.9)
2	183 (24.9)
3	315(42.9)
4	189 (25.7)
Autotransplanted parathyroids	30 (4.0)
Postoperative hypocalcemia	231 (31.4)
Definitive hypocalcemia	17 (2.3)

Legend: TNM: Tumor Nodes Metastasis.

**Table 2 jcm-12-01956-t002:** Risk factors for low levels of postoperative PTH.

Characteristics	PTH < 12 pg/mL *N* (%)	PTH > 12 pg/mL *N* (%)	*p*-Value
Patients	230 (31.3)	504 (68.7)	
Age (mean, years)	51.5 ± 11.3	58.2 ± 13.2	**0.012**
Age groups			
<40	46 (20)	89 (17.6)	**<0.05**
41–55	88 (38.2)	160 (31.7)	0.447
>55	96 (41.8)	255 (50.5)	0.632
Sex			
Male	34 (14.8)	127 (25)	
Female	196 (85.2)	377 (75)	**0.001**
Surgical Procedure			
Total thyroidectomy	230 (100)	474 (94)	0.885
Lobectomy	0	32 (6)	
Neck dissection	80 (35)	130 (26)	**<0.05**
Central neck dissection	36 (15.6)	56 (15.3)	0.756
Unilateral Lymphadenectomy	35 (15.2)	60 (11.9)	0.447
Bilateral lymphadenectomy	9 (3.9)	14 (2.7)	0.322
PTH levels (mean)			
Preoperative	52.7 ± 25.9	54.1 ± 21.1	0.440
Postoperative	4.8 ± 3.73	37.0 ± 17.6	**<0.001**
Incidental parathyroidectomy			
No	168 (73)	444 (87.9)	
Yes	61 (27)	61 (12.1)	**<0.001**
Autotransplanted parathyroids	18 (7.8)	12 (2.3)	0.224
Parathyroid glands identified during surgery			
0	5 (2.1)	6 (1.2)	0.845
1	8 (3.4)	28 (5.5)	0.748
2	60 (26)	123 (24.4)	0.732
3	86 (37.3)	229 (45.4)	0.196
4	71 (31.2)	118 (23.5)	0.242
Diagnosis			
Cancer	60 (26)	118 (23.4)	0.434
Lymph node metastases	12 (5.2)	11 (2.1)	**0.028**
Multinodular goiter	145 (63)	332 (65.9)	0.453
Hashimoto thyroiditis (HT)	27 (12)	79 (15.7)	0.375
Cancer + HT	7 (3)	28 (5.5)	0.138
Thyroid volume (mean, cm^2^)	35.2 ± 23.71	33.1 ± 21.8	0.675
Number of retrieved lymph nodes (mean)	8.30	4.09	**0.005**
Postoperative hypocalcemia			
Temporary	147 (64)	84 (16.7)	**<0.001**
Definitive	17 (7)	0	
Preoperative/postoperative PTH levels ratio	90.9%	29.1%	**<0.001**
6-month postoperative serum calcium (mean, g/dL)	8.6 ± 0.48	9.1 ± 0.44	0.587
6-month postoperative PTH level (mean, pg/mL)	17.15 ± 8.26	25.1 ± 9.32	0.201

Legend: PTH: parathyroid hormone; bold character in *p* column indicates a statistical significance.

**Table 3 jcm-12-01956-t003:** Multivariate analysis for predictive factors associated with temporary hypocalcemia.

Characteristics	OR	95% CI	*p*-Value
Sex			
Male	1		
Female	1.91	1.281–2.943	**0.001**
PTH < 12 pg/mL	3.71	2.592–5.322	**<0.001**
Neck dissection	1.44	0.840–2.490	0.186
Incidental parathyroidectomy	1.49	0.970–2.302	**0.049**

Legend: PTH: parathyroid hormone; bold character in *p* column indicates a statistical significance.

**Table 4 jcm-12-01956-t004:** Risk factors for unintentional parathyroidectomy.

Characteristics	Unintentional Parathyroidectomy	No Parathyroidectomy	
	*N* (%)	*N* (%)	*p-*Value
Patients	122 (16.6)	612 (83.4)	
Sex			
Male	24 (19.6)	137 (22.3)	0.532
Female	98 (79.4)	466 (77.7)	0.554
Age (years, %)	51.1 ± 15.5	53.8 ± 12.8	< 0.05
<55	76 (62.2)	325 (53.1)	
>55	46 (37.8)	287 (46.9)	0.435
Age Groups			
<40	32 (26.2)	105 (17.1)	**0.014**
41–55	48 (39.3)	302 (49.3)	0.432
>55	42 (34.4)	207 (33.8)	0.693
Total thyroidectomy/lobectomy	117 (95.9)	584 (95.4)	0.765
Lobectomy	5 (4.1)	28 (4.6)	0.771
Neck dissection	52 (42.6)	158 (25.7)	**<0.001**
Number of retrieved lymph nodes			
<4	17	32	OR 2.93 (95% CI 1.57–5.47, *p* < 0.05)
5–8	1	10	OR 0.4 (95% CI 0.06–3.92, *p* = 0.496)
>8	7	11	OR 3.3 (95% CI 1.26–8.75, *p* = 0.010)
Preoperative PTH (mean, pg/mL)	55 ± 26.6	53.4 ± 22.6	0.356
Postoperative PTH (mean, pg/mL)	18.1 ± 20.0	29.2 ± 23	**<0.001**
PTH < 12 pg/mL	64 (52.4)	168 (27.4)	**<0.001**
Temporary hypocalcemia < 8 mg/dL	60 (49.1)	171 (27.9)	**0.024**
Definitive hypocalcemia	6 (4.9)	11 (1.7)	**0.036**
Underlying disease			
Cancer	42 (34.4)	145 (23.7)	**0.013**
Multinodular goiter	71 (58.2)	439 (71.6)	0.435
Hashimoto thyroiditis	17 (13.9)	94 (15.3)	0.654
Others	2 (1.6)	4 (1.6)	0.553
Thyroid volume (mean, cm^2^)	26.0 ± 23	36.4 ± 32.8	**0.006**
Postoperative serum calcium (mean, mg/dL)	8.1 ± 0.65	8.2 ± 0.6	0.07
6-month postoperative serum calcium (mean, g/dL)	8.9 ± 0.58	9.4 ± 0.42	0.838

Legend: PTH: parathyroid hormone; bold character in *p* column indicates a statistical significance.

**Table 5 jcm-12-01956-t005:** Multivariate analysis for predictive factors of temporary hypocalcemia in patients with incidental parathyroidectomy.

Characteristics	OR	95% CI	*p*-Value
Age			
>40 y	1		
<40 y	1.74	0.933–3.280	**0.043**
PTH < 12 pg/mL	1.66	0.933–2.959	**0.048**
Neck dissection	2.40	1.115–5.223	**0.022**
Thyroid cancer	1.93	1.038–3.614	**0.035**

Legend: PTH: parathyroid hormone; bold character in *p* column indicates a statistical significance.

## Data Availability

De-identified data can be made available upon reasonable request.

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
