# Peer review of "Risk Factors for Hypoparathyroidism after Thyroid Surgery: A Single-Center Study"

_jcm, 2023, doi:10.3390/jcm12051956_

Round 1
Reviewer 1 Report
The article entitled “Risk factors for hypoparathyroidism after thyroid surgery: a single center study”by Privitera et al. analyzes a very large series of patients and assesses their risk factors of postoperative hypocalcemia.
The authors include 734 patients of whom 702 patients underwent total thyroidectomy and 32 lobectomy. They found 31.4% (231 patients) of transient and 2.3% (17 patients) of definitive hypocalcemia.
Patients are divided into two groups according to PTH levels (< or >12 pg/ml.)
In the group of patients with PTH <12 pg/ml (230; 31.3%), the authors found 64% (147 patients) of transient hypocalcemia.
In the group with PTH >12 pg/ml transient hypocalcemia was 16.7% (84 patients). 17 patients (7%) with PTH <12 pg/ml had definitive hypocalcemia.
The highlight of this analysis is the very large number of the sample analyzed and the work is very interesting.
However, there are points that need to be clarified and discussed in detail.
- The authors include lobectomies in the analysis
I believe that this group, although small (32 patients), should be removed from the analysis as a cause of bias.
Lobectomy alone cannot be considered at risk for hypoparathyroidism since the two glands on the preserved side have not been exposed to risk and therefore it is also superfluous to dose calcium on postoperative Day 1.
-Why did the authors take 12 pg/ml as the PTH cut off?
The authors should explain why they preferred to use this cut off in materials and methods.
-Why do the authors analyze calcemia and PTH separately?
the two data should be correlated.
This point should be better clarified in the methods.
It should also be explained whether corrected calcemia is taken into account or not.
There are numerous articles on the subject, I recommend that the authors read these recent publications on the subject:
-Predicting risk factors of postoperative hypocalcemia after total thyroidectomy: is safe discharge without supplementation possible? A large cohort study. Langenbecks Arch Surg. 2021 Nov;406(7):2425-2431. doi: 10.1007/s00423-021-02237-2. Epub 2021 Aug 10.
- Luo H, Yabg H, Wei T et al (2017). Protocol for management after thyroidectomy: a retrospective study based on one-center experience. Ther Clin Risk Manag. 15; 13: 635-641.
-Saba A, Podda M, Messina Campanella A et al (2017). Early prediction of hypocalcemia following thyroid surgery. A prospective randomized clinical trial. Langenbecks Arch Surg 402 (7): 1119-1125.
-The authors also took into account the type of surgery performed, cases of accidental parathyroidectomy, and the presence of thyroiditis. All data are shown in Table 2.
To better enhance this work, I recommend that they better specify these data point by point in the results.
-Line 41-42: “most frequent complication after thyroid surgery”.
Hypocalcemia is a common complication after total thyroidectomy with or without lymphadenectomy; there is no risk in case of lobectomy or isthmectomy.
Therefore, I recommend that the authors add "total thyroidectomy” instead thyroid surgery. I recommend also removing “undergoing thyroid surgery” which in my opinion is a repetition.
-Line 47-50: “Hypocalcemia after ….supplementation”.
Please reformulate this sentence.
-Line 90-91: “underwent a total…a lobectomy”
Please remove “a”.
-Line 93: “no patient …convulsions”.
Convulsions are very rare.... maybe the authors meant that the patients had no clinical signs?
I think it's better to rephrase it.
-In the table 1, “13 patients had lymph node metastasis”.
In the materials and methods, the authors state that lymphadenectomy was performed when necessary. They should indicate when it was performed and include in the results whether it was done.
Central neck dissection is a cause of hypoparathyroidism and should not be underestimated.
I recommend that the authors read this recent publication on the subject:
Surgical complications in prophylactic central neck dissection: preliminary findings from a retrospective cohort study. Minerva Chir 69 (suppl1, n°2): 131-4.
-Lines 103-104, in the Results they find an interesting result: “patients with lymph-node metastasis were at higher risk…..”
I recommend that this point be better described in the materials and methods.
-The authors find that female sex is a risk factor but this is an obvious conclusion being the female sex the most represented and being the woman most susceptible to thyroid pathology. I think it is better to specify it.
I believe that such changes can improve the scientific value of this article which remains very interesting.
Author Response
The article entitled “Risk factors for hypoparathyroidism after thyroid surgery: a single center study”by Privitera et al. analyzes a very large series of patients and assesses their risk factors of postoperative hypocalcemia.
The authors include 734 patients of whom 702 patients underwent total thyroidectomy and 32 lobectomy. They found 31.4% (231 patients) of transient and 2.3% (17 patients) of definitive hypocalcemia.
Patients are divided into two groups according to PTH levels (< or >12 pg/ml.)
In the group of patients with PTH <12 pg/ml (230; 31.3%), the authors found 64% (147 patients) of transient hypocalcemia.
In the group with PTH >12 pg/ml transient hypocalcemia was 16.7% (84 patients). 17 patients (7%) with PTH <12 pg/ml had definitive hypocalcemia.
The highlight of this analysis is the very large number of the sample analyzed and the work is very interesting.
However, there are points that need to be clarified and discussed in detail.
- The authors include lobectomies in the analysis
I believe that this group, although small (32 patients), should be removed from the analysis as a cause of bias.
Lobectomy alone cannot be considered at risk for hypoparathyroidism since the two glands on the preserved side have not been exposed to risk and therefore it is also superfluous to dose calcium on postoperative Day 1.
I completely agree with you that the evaluation of hypocalcemia in patients undergoing lobectomy may be not relevant. However, patients undergoing lobectomy have been included for the evaluation of the rate of incidental parathyroidectomy and, for completeness of data evaluation, they have been included also in the analysis of hypocalcemia.
-Why did the authors take 12 pg/ml as the PTH cut off?
The authors should explain why they preferred to use this cut off in materials and methods.
This was reported in material and methods
-Why do the authors analyze calcemia and PTH separately?
the two data should be correlated.
This point should be better clarified in the methods.
It should also be explained whether corrected calcemia is taken into account or not.
There are numerous articles on the subject, I recommend that the authors read these recent publications on the subject:
-Predicting risk factors of postoperative hypocalcemia after total thyroidectomy: is safe discharge without supplementation possible? A large cohort study. Langenbecks Arch Surg. 2021 Nov;406(7):2425-2431. doi: 10.1007/s00423-021-02237-2. Epub 2021 Aug 10.
- Luo H, Yabg H, Wei T et al (2017). Protocol for management after thyroidectomy: a retrospective study based on one-center experience. Ther Clin Risk Manag. 15; 13: 635-641.
-Saba A, Podda M, Messina Campanella A et al (2017). Early prediction of hypocalcemia following thyroid surgery. A prospective randomized clinical trial. Langenbecks Arch Surg 402 (7): 1119-1125.
Thank you for your suggestions. PTH levels and calcemia were both measured 6 hours after surgery, and calcemia also 24 hours after surgery The suggested reference has been inserted in the manuscript
-The authors also took into account the type of surgery performed, cases of accidental parathyroidectomy, and the presence of thyroiditis. All data are shown in Table 2.
To better enhance this work, I recommend that they better specify these data point by point in the results.
Data were reported more extensively
-Line 41-42: “most frequent complication after thyroid surgery”.
Hypocalcemia is a common complication after total thyroidectomy with or without lymphadenectomy; there is no risk in case of lobectomy or isthmectomy.
Therefore, I recommend that the authors add "total thyroidectomy” instead thyroid surgery. I recommend also removing “undergoing thyroid surgery” which in my opinion is a repetition.
Thank you for your suggestion. The text was revised accordingly.
-Line 47-50: “Hypocalcemia after ….supplementation”.
Please reformulate this sentence.
Thank you for your suggestion. The text was modified accordingly
-Line 90-91: “underwent a total…a lobectomy”
Please remove “a”.
The text was modified accordingly
-Line 93: “no patient …convulsions”.
Convulsions are very rare.... maybe the authors meant that the patients had no clinical signs?
I think it's better to rephrase it.
The text was modified accordingly
-In the table 1, “13 patients had lymph node metastasis”.
In the materials and methods, the authors state that lymphadenectomy was performed when necessary. They should indicate when it was performed and include in the results whether it was done.
Central neck dissection is a cause of hypoparathyroidism and should not be underestimated.
I recommend that the authors read this recent publication on the subject:
Surgical complications in prophylactic central neck dissection: preliminary findings from a retrospective cohort study. Minerva Chir 69 (suppl1, n°2): 131-4.
-Lines 103-104, in the Results they find an interesting result: “patients with lymph-node metastasis were at higher risk…..”
I recommend that this point be better described in the materials and methods.
Thank you very much for your suggestions. Unfortunately, to avoid similarity with our previous manuscript, we had to remove many sentences from material and methods. However, we tried to specify better when and how lymphadenectomy was performed
-The authors find that female sex is a risk factor but this is an obvious conclusion being the female sex the most represented and being the woman most susceptible to thyroid pathology. I think it is better to specify it.
A brief comment was added to discussion
I believe that such changes can improve the scientific value of this article which remains very interesting.
Thank you very much for your relevant and significant suggestions that will improve significantly the scientific value of the manuscript
Reviewer 2 Report
This has been an extensively explored topic, and there is already a systematic review and meta-analysis present on this topic. An additional single-center study is unlikely to yield much additional perspective. If the authors can justify this study, in the face of already existing similar studies, then this needs to be clearly mentioned to the reader in the Introduction and the Discussion. Currently, there is no mention of the existing meta-analysis.
In addition, the statistical analysis is extremely basic, which further reduces the value of the article and its conclusion. With a sample size in excess of 700, the authors should definitely try to execute some level of adjusted analysis. They can simply conduct multivariable regression to report adjusted odds ratios in place of the current crude odds ratios reported.
Author Response
This has been an extensively explored topic, and there is already a systematic review and meta-analysis present on this topic. An additional single-center study is unlikely to yield much additional perspective. If the authors can justify this study, in the face of already existing similar studies, then this needs to be clearly mentioned to the reader in the Introduction and the Discussion. Currently, there is no mention of the existing meta-analysis.
Although I agree with the reviewer that the topic has been extensively explored in literature, I believe that this manuscript could be of interest for readers since it involves one of the largest series reported in literature and also presents the incidence and risk factors for incidental parathyroidectomy, which have been rarely reported in literature. Current meta-analysis and review articles were included in the manuscript as suggested.
In addition, the statistical analysis is extremely basic, which further reduces the value of the article and its conclusion. With a sample size in excess of 700, the authors should definitely try to execute some level of adjusted analysis. They can simply conduct multivariable regression to report adjusted odds ratios in place of the current crude odds ratios reported.
Thank you for your suggestions. However there are two multivariate analyses in the manuscript (Table 3 and 5): however, this was better specified in the material and methods section .